# Proteogenomic Analysis of *Burkholderia* Species Strains 25 and 46 Isolated from Uraniferous Soils Reveals Multiple Mechanisms to Cope with Uranium Stress

**DOI:** 10.3390/cells7120269

**Published:** 2018-12-12

**Authors:** Meenakshi Agarwal, Ashish Pathak, Rajesh Singh Rathore, Om Prakash, Rakesh Singh, Rajneesh Jaswal, John Seaman, Ashvini Chauhan

**Affiliations:** 1Environmental Biotechnology Laboratory, School of the Environment, 1515 S. Martin Luther King Jr. Blvd., Suite 305B, FSH Science Research Center, Florida A&M University, Tallahassee, FL 32307, USA; meenakshiagarwal.iitb@gmail.com (M.A.); ashishpathak72@gmail.com (A.P.); rajeshrathore854@gmail.com (R.S.R.); jaswal.rajneesh@gmail.com (R.J.); 2National Centre for Microbial Resource, National Centre for Cell Science, Pune 411007, India; prakas1974@gmail.com; 3Translational Science Lab, College of Medicine, Florida State University, Tallahassee, FL 32304, USA; rakesh.singh@med.fsu.edu; 4Savannah River Ecology Laboratory, University of Georgia, Aiken, SC 29802, USA; seaman@srel.uga.edu

**Keywords:** genomics, proteomics, uranium, *Burkholderia*

## Abstract

Two *Burkholderia* spp. (strains SRS-25 and SRS-46) were isolated from high concentrations of uranium (U) from the U.S. Department of Energy (DOE)-managed Savannah River Site (SRS). SRS contains soil gradients that remain co-contaminated by heavy metals from previous nuclear weapons production activities. Uranium (U) is one of the dominant contaminants within the SRS impacted soils, which can be microbially transformed into less toxic forms. We established microcosms containing strains SRS-25 and SRS-46 spiked with U and evaluated the microbially-mediated depletion with concomitant genomic and proteomic analysis. Both strains showed a rapid depletion of U; draft genome sequences revealed SRS-25 genome to be of approximately 8,152,324 bp, a G + C content of 66.5, containing a total 7604 coding sequences with 77 total RNA genes. Similarly, strain SRS-46 contained a genome size of 8,587,429 bp with a G + C content of 67.1, 7895 coding sequences, with 73 total RNA genes, respectively. An in-depth, genome-wide comparisons between strains 25, 46 and a previously isolated strain from our research (*Burkholderia* sp. strain SRS-W-2-2016), revealed a common pool of 3128 genes; many were found to be homologues to previously characterized metal resistance genes (e.g., for cadmium, cobalt, and zinc), as well as for transporter, stress/detoxification, cytochromes, and drug resistance functions. Furthermore, proteomic analysis of strains with or without U stress, revealed the increased expression of 34 proteins from strain SRS-25 and 52 proteins from strain SRS-46; similar to the genomic analyses, many of these proteins have previously been shown to function in stress response, DNA repair, protein biosynthesis and metabolism. Overall, this comparative proteogenomics study confirms the repertoire of metabolic and stress response functions likely rendering the ecological competitiveness to the isolated strains for colonization and survival in the heavy metals contaminated SRS soil habitat.

## 1. Introduction

Savannah River Site (SRS) located in Aiken, SC, functioned as a nuclear materials production facility for the U.S. Department of Energy (DOE), where metal-clad uranium (U) targets were used in the production of plutonium [1]. From 1954 to 1982, SRS wastewater originating from the metal plating and fabrication processes was released directly into Tims Branch, a second-order stream. Because of these activities, large quantities of U (depleted as well as naturally-occurring) was released and deposited into stream sediments and an abandoned farm pond—the Steeds Pond, that served as a natural settling basin along Tims Branch [1,2]. Although contaminant distribution is heterogeneous in the Steeds Pond sediments, U concentration is found upwards of 1000 mg/kg [2]. To gauge in-situ remediation (natural attenuation) and recommend appropriate strategies for the remediation of contaminated SRS ecosystems, it is necessary to obtain a deeper understanding of the mechanisms possessed by the native soil microbiota to resist metals and radionuclides. In fact, microorganisms exposed to the contaminant stresses not only have the unparalleled ability to survive in radionuclide contaminated environments but also potentially reduce toxicity of U [3,4,5]. For example, 90% of bacteria belonging to *Firmicutes, Gammaproteobacteria, Actinobacteria, Bacteroidetes* and *Betaproteobacteria* were found to resist high uranium concentration of up to 4 mM from a uranium mining site in Domiastat, India [6]. Kulkarni et al. isolated two phosphatase producing bacteria- *E. coli* and *Deinococcus radiodurans*, and showed that both grew in the presence of up to 20 mM uranium [7]. Evolutionary mechanisms that include bioreduction, biosorption, biomineralization, and bioaccumulation are utilized by environmental microbiota to persist in uraniferous niches [4]. Among these mechanisms, U biomineralization has recently garnered significant interest because this bioremediative process entails formation of U(VI)phosphate minerals as a function of microbial enzymatic activities and consequently, U is sequestered as stable solid mineral phases within the environmental matrices [8,9]. However, genomic mechanisms underpinning U transformation into the less-mobile and less-toxic forms, such as a U mineral, continue to remain understudied and hence unclear, especially for the SRS uraniferous soils.

Soil samples for this study were collected from site 101 located within the Tims Branch system. At this site, U concentrations are typically present between 700–800 ppm [5], which is a high concentration based on a previously reported criteria [10]. Therefore, the historical contamination of the SRS site presents with an opportunity to study genome-enabled mechanisms recruited by the SRS-native microorganisms facilitating their survival in co-contaminated environments. In fact, the stress posed by environmental contaminants facilitate recruitment of genes by horizontal gene transfer mechanisms that enable the microbial cells to not only resist but also bioremediate the contaminants, mostly by the synthesis of proteins for cellular survival [11]. Some examples of such cellular and genomic-mechanisms include membrane-bound efflux pumps, presence of metal resistant genes, detoxification genes, and biosorption/bioaccumulation of the contaminant at or within the bacterial cell membrane [12].

To further understand environmentally-relevant molecular mechanisms that underpin microbial survival in radionuclide and heavy metal-rich ecosystems, we recently isolated several bacterial strains in the presence of high uranium concentrations [13]. 16S rRNA gene sequence based analysis revealed that the isolated strains mainly belonged to *Burkholderia* spp. and *Arthrobacter* spp., these genera have been previously demonstrated to serve as bioindicators of environmental contamination as well as agents of bioremediation, especially U [14]. Notably, *Burkholderia* spp., have also been shown by others to successfully thrive in habitats that pose extreme environmental stresses, including uranium-rich ecosystems. For example, *Burkholderia*-like microorganisms comprised a significant proportion of the soil microbiota at the Integrated Field-Scale Subsurface Research Challenge Site (IFRC), Oak Ridge, Tennessee [15] as well as Rifle, Colorado [16]. The intrinsic ability of Burkholderiales to dominate uraniferous habitats is not entirely surprising, owing in part to their repertoire of metabolic abilities, attributed to the large, multi-replicon genome, conferring genome plasticity and metabolic versatility [17], respectively. Adding to this knowledge on Burkholderiales, our recent study also showed genome-enabled mechanisms possessed by another soil isolate—*Burkholderia* sp. strain SRS-W-2-2016, that colonized the heavy metal-rich SRS soil habitat [13]. From these studies, it was evident that *Burkholderia* spp. recruited a suite of genomic traits to facilitate survival in radionuclide and metal-contaminated habitats. Such traits included several gene homologues previously demonstrated to render resistance against heavy metals and radionuclides, including a suite of substrate-binding proteins, permeases, transport regulators, and efflux pumps. These likely work in concert to potentially detoxify toxic metals and thus facilitate the natural attenuation of contaminants within the SRS-impacted ecosystem. Another interesting genomic trait that our previous study identified in the *Burkholderia* sp. strain SRS-W-2-2016 was the presence of several genomic islands (GEIs). Notably, bacterial genomes not only consist of the “core” set of genes that provide for essential metabolic functions but can also harbor genes acquired from the bacterium’s native environment via horizontal gene transfer (HGT) mechanisms. These “foreign” genes, typically occurring as orthologous genomic blocks, called genomic islands [18], can provide several beneficial traits to the host bacterium, including environmental adaptations and genomic plasticity. In fact, GEIs have been functionally classified into the following four broad categories: pathogenicity islands (PAIs), harboring virulence genes; metabolic islands (MIs), genes that code for secondary metabolite biosynthesis; resistance islands (RIs), genes that typically provide resistance against antibiotics; and symbiotic islands (SIs), those genes that facilitate symbiotic associations of the host with other micro- and macroorganisms, respectively. Thus, genome-wide mining of environmentally relevant microorganisms (e.g., Burkholderiales), can provide a broader understanding on the basis of metal-*Burkholderia* interactions, which can form the basis of targeted management of desirable microbial traits, including bioremediation, resulting in better stewardship of nuclear-legacy contaminated environments.

Note, however that, to obtain a holistic understanding at the cellular level of bacteria, against uranium, analysis that combines the genomics and proteomics dataset, termed as proteogenomics, can be a very powerful approach. Such an approach has the ability to provide a unique peek into the uranium-microbe interaction(s), including other relevant metabolic and functional traits possessed by the bacteria under study. Despite this, only a few studies are available on the impacts of U stress at the proteomic level, especially for aerobic microbiota. To further advance on this aspect of metal-microbe interactions, Gallois and coworkers recently integrated the genomics and proteomics data of *Microbacterium oleivorans* A9, a uranium-tolerant actinobacteria isolated from near the Chernobyl nuclear power plant, when grown in the presence or absence of uranyl nitrate [19]. This study revealed 1532 proteins and specifically, 591 proteins with significant differences in abundances when gown with or without uranium. Under the influence of U(VI) exposure, Yung et al., identified a phytase enzyme and an ABC transporter to be upregulated in *Caulobacter crescentus* [20], which can tolerate high concentrations of U [21]. Therefore, the objective of this study was to obtain genomics and high-throughput proteomics data on strains SRS-25 and SRS-46, grown with and without uranium, thus forming a strong basis of our understanding on the underpinnings of U-*Burkholderia* interactions, which to our understanding, has not been previously demonstrated. Such studies can lead to the identification of gene-protein targets to not only gauge but predict the trajectory of uranium bioremediation in historically U-contaminated environments.

## 2. Material and Methods

### 2.1. Isolation and U Resistance Studies on Strains SRS-25 and SRS-46

*Burkholderia* sp. 25 and 46 were isolated from uraniferous SRS soils collected from the Tims Branch/Steeds Pond area. Briefly, soil samples were serially diluted and 100 μL of this slurry was plated onto LB agar supplemented with uranyl nitrate at a concentration of 4.2 mM; this mimics the uranium concentration present in-situ within the SRS soils [22]. Cells that grew on LB + U media were also screened on Tryptose Phosphate Methyl Green (TPMG) media to screen for phosphatase-based uranium biomineralization activity as shown before [22,23]. Two of the most robustly growing isolates on uranium with phosphatase positive activity on TPMG media-named as strains #25 and #46 were selected for further physiological and proteogenomic studies.

Resistance of strains 25 and 46 against different concentrations of uranium was evaluated by growth, which was analyzed using the Bioscreen C system (Growth Curves USA, Piscataway, NJ, USA), as reported previously [22]. Briefly, 4 M media [24] (modified with the addition of 0.04% yeast extract) was supplemented with U ranging from 0–5000 µM. The assay was run using the honeycomb Bioscreen C plates containing 290 µL medium and 10 µL of the inoculum, which was grown overnight to an OD_600_ of 0.3 ± 0.05. The instrument was programmed to perform regular shaking and capture OD_600_ at increments of every 3 h for 68 h; these experiments were run in triplicates and averaged values are reported.

### 2.2. Uranium Depletion by Strains SRS-25 and SRS-46

To determine U remediation potential of *Burkholderia* sp. strains 25 and 46, microcosms were established in 4 M media supplemented with 1000 µM U to which were then added overnight grown cultures to a final OD_600_ of 0.05 ± 0.01, as reported recently [13,22]. Briefly, flasks were incubated at 30 °C and 120 rpm and samples were taken every 24 h for 2 days. The samples were centrifuged, and supernatant was acidified using HNO_3_ to a final concentration of 2%. Uranium depletion in the supernatants was measured by inductively coupled plasma-mass spectrometry (ICP-MS) on a NexIon 300 (Perkin Elmer, Waltham, MA, USA) in accordance with the quality assurance (QA) and quality control (QC) protocols of EPA method 6020B (USEPA, 2014; Method 6020B, Rev. 2. Inductively coupled plasma-mass spectrometry (ICP-MS), Office of Solid Waste, Washington, DC, USA). Cell pellets were saved immediately in −80 °C collected at time 0, day 1 and day 2 for proteomics studies.

### 2.3. Genomic Characterization of Strains SRS-25 and SRS-46

Genomic DNA from strains SRS-25 and SRS-46 was extracted and prepared for sequencing on an Illumina HiSeq2000 instrument, as described previously [13]. De novo assembly of the raw reads was performed with the SPAdes assembler [25] using default settings. Assembly coverage statistics were computed by mapping raw reads to the assembled genome using bowtie2 [26]. Specifically, we determined a coverage filter for the contigs from the distribution of coverage levels across the assembly. First, contigs were ordered by coverage, and cumulative assembly length was computed across all contigs. Coverage level at 50% of the total cumulative assembly length was determined and half of that coverage level was selected as a coverage filter. The remaining reads were aligned with nucmer [25] against the closest reference sequence from NCBI (determined by a BLAST of the 16S rRNA sequence): accession numbers CP002013.1, CP002014.1, and NC_014119.1 for chromosomes 1, 2, and 3. All contigs, for each strain, were aligned to these references, and the optimal contig ordering and orientation to most closely match the reference was determined using mummerplot [25] with layout specified. Contigs were then reordered and reversed as needed to match the ordering determined by mummerplot. Circular genomic maps were generated using the CGView Comparison Tool [27]. 

The genomes, with a coverage of 250x, were then annotated and genes predicted by IMG/er [28], RAST [29] and NCBI’s Prokaryotic Genomes Automatic Annotation Pipeline (PGAAP), version 2.0. Genome-based phylogenetic tree of the strains were constructed using the One Codex database platform [30], which generates taxonomic classification of nucleotide reads by assessing for exact k-mer matches against their database of bacterial, viral, and fungal genomes as well as the National Center for Biotechnology Information’s (NCBI) Reference Sequence Database. Comparative genomics of strains SRS-25 and SRS-46 relative to closest taxonomic relatives was performed by EDGAR [31].

### 2.4. Protein Extraction and Separation

Protein extraction and mass spectrometry were performed at Translational Science Laboratory, Florida State University. Cells (50) mg were collected from treated (with uranyl nitrate) and untreated microcosms from the day 1 time point, which coincided with the maximum depletion of uranium, followed by resuspension in 500 µL of SDT lysis buffer (4% SDS, 20 mM Tris-HCl (pH 8.6), 50 mM DTT, protease inhibitor tablets (ThermoFisher Scientific, Gainesville, FL, USA), and MS grade water). Suspended cells were frozen and thawed three times in liquid nitrogen and at 95 °C hot bath respectively for 15 min. Approximtely 50 mg of glass beads (G8772-Sigma 425–600 µm) were added to the cell suspension and subjected to 10 cycles of bead beating for each 45 s cycle and 3 min ice interval between every cycle. Lysed cells were centrifuged at 16,000 g for 15 min and supernatant was collected in the fresh tube. Total protein was quantified by Pierce^TM^ BCA Protein Assay kit (Thermo Scientific, Waltham, MA, USA) using a Nanodrop Spectrophotometer (Thermo Scientific, Waltham, MA, USA).

For protein separation on SDS-PAGE, 30 µg of protein was loaded in precast-mini Protean R TFX^TM^ Gels (4–20%, 10 well combs, 50 mL BioRad, Hercules, CA, USA) and stained with Coomassie blue dye. Samples were fractionated by cutting the gel lanes into four sections from destained gel. Individual gel pieces were kept into a separate eppendorf tubes. In-gel digest was performed using ProteoExtract All-in-One Trypsin Digestion Kit (Cat. No. 650212 Calbiochem, San Diego, CA, USA) according to manufacturer’s instructions. Briefly, carefully excised gel pieces were destained with wash buffer, dried at 95 °C for 15 min. Gels were rehydrated with digest buffer, reduced using the reducing agent for 10 min at 37 °C. Samples were cooled to RT and then blocked using blocking reagent for 10 min at room temperature. Trypsin at a final concentration of 8 ng/µL was added and incubated for 2 h at 37 °C with shaking. Peptides were eluted in 50 μL 0.1% FA and run on LCMS.

### 2.5. Mass Spectrometry and Protein Identification

An externally calibrated Thermo Q Exactive HF (high-resolution electrospray tandem mass spectrometer) was used in conjunction with Dionex UltiMate3000 RSLC nano System (Thermo Scientific, Waltham, MA, USA). A 5 μL sample was aspirated into a 50 μL loop and loaded onto the trap column (Thermo µ-Precolumn 5 mm, with nanoViper tubing 30 µm i.d. × 10 cm). The flow rate was set to 300 nL/min for separation on the analytical column (Acclaim pepmap RSLC 75 μM × 15 cm nanoviper, Thermo Scientific, Waltham, MA, USA). Mobile phase A was composed of 99.9 H_2_O (EMD Omni Solvent, Millipore Sigman, Austin, TX, USA), and 0.1% formic acid and mobile phase B was composed of 99.9% ACN, and 0.1% formic acid. A 60 min linear gradient from 3% to 45% B was performed. The LC eluent was directly nanosprayed into Q Exactive HF mass-spectrometer (Thermo Scientific). During the chromatographic separation, the Q-Exactive HF was operated in a data-dependent mode and under direct control of the Thermo Excalibur 3.1.66 (Thermo Scientific). The MS data were acquired using the following parameters: 20 data-dependent collisional-induced-dissociation (CID) MS/MS scans per full scan (350 to 1700 *m*/*z*) at 60,000 resolution. MS2 were acquired in centroid mode at 15,000 resolution. Ions with single charge or charges more than 7 as well as unassigned charge were excluded. A 15 s dynamic exclusion window was used. All measurements were performed at RT and three technical replicates were run for each sample. The raw files were analyzed using Proteome Discoverer (version 2.0, ThermoFisher Scientific, Gainesville, FL, USA) software package with SequestHT and Mascot search nodes using species specific tremble fasta database and the Percolator peptide validator. The resulting msf files were further analyzed by the proteome validator software ‘Scaffold version 4.4′ (Portland, OR, USA).

### 2.6. Statistical Analysis

Statistical analysis on the genomic data was performed using the embedded tools within the pipelines used for comparative genomics. The proteomics data sets were combined as a union, imputing abundance values of 0 if a protein was found in one data set but not in the other. Ordination analyses were then performed using non-metric multidimensional scaling (NMDS) with the Bray-Curtis similarity coefficient in Primer-E version 6.1.13 and PERMANOVA version 1.0.3 (Albany, Auckland, New Zealand). Also computed were the log2 fold-change between control and uranium samples in each genome, and the top 50 proteins based on the maximum log2 fold-change in either genome were plotted as a heatmap.A complete-linkage hierarchical clustering on proteins and across samples were run and plotted as heatmaps.

### 2.7. Genomic and Proteomic Data Accession Numbers

The Whole Genome Shotgun of *Burkholderia* species reported in this study have been deposited at DDBJ/ENA/GenBank under BioSample: SAMN08567921 (SRS-25); SAMN08567954 (SRS-46) and SAMN06141630 (SRS-W-2-2016). The mass spectrometry proteomics data have been deposited to the ProteomeXchange Consortium via the PRIDE [32] repository with the dataset identifier PXD011367 and 10.6019/PXD011367.

## 3. Results

### 3.1. Depletion of Uranium by Strains SRS-25 and SRS-46

Two robustly U-resistant strains, named SRS-25 and SRS 46, were isolated from a uraniferous Savannah River Site (SRS) soils. The phosphatase-based U biomineralization activity of these strains was checked on TPMG media as previously shown [22,23], which showed a phosphatase positive response (please see graphical abstract). To determine the U resistance potential of strains SRS-25 and SRS-46 different concentrations of U were supplemented in 4 M media and growth results are presented in Figure 1A,B. Strains SRS-25 and 46 could resist up to 5000 µM of U, which is the typical concentrations of U in the sampling location [13], although with an increased lag phase. Rapid growth occurred at the lower concentration of 1000 µM U and stationary phase was reached in 24 h (Figure 1A,B). This suggested that 1000 µM of U is likely one of the best concentrations to run proteomic analysis on strains SRS-25 and SRS-46, respectively.

The above resistance abilities of strains SRS-25 and SRS-46 to U were further confirmed by ICP-MS analysis on microcosms established with 1000 µM U followed by quantifying the depletion of amended U; these results are shown in Figure 1C. Notably, the concentration of U rapidly declined in merely 24 h; strongly suggesting that both strains possessed the ability to rapidly biotransform and/or bioremediate U. However, in both strains, more specifically in strain 25, uranium started to reappear after being completely depleted from the supernatant (Figure 1C); it is likely that some fraction of uranium was biosorbed onto the cellular surface after immediate exposure to uranium and eventually the cellular-bound uranium began to desorb into the supernatant as a function of time. It could also be that initially U was transported into the cells and by 33 h post-exposure, efflux pumps started to dump some of this uranium extracellularly.

### 3.2. Genome-Centric Evaluation to Identify Metal Resistance Basis of Isolated Strains

The whole genome sequence of strain SRS-25 assembled into 66 contigs with an N50 of 285073 and 23 contigs with an N50 of 662813 for SRS-46, respectively. As shown in Figure 2A,B, strain SRS-25 possessed a genome of approximately 8,152,324 bp, a G + C content of 66.5, containing a total 7604 coding sequences with 77 total RNA genes. Similarly, strain SRS-46 was predicted to be approximately 8,587,429 with a G + C content of 67.1 and contained 7895 coding sequences, with 73 total RNA genes, respectively. The genomic traits of strains SRS-25 and SRS-46 stated above are in line with the *Burkholderia* genomes, which are known to be variable in size ranging from 2.4 Mb (Ca. *Burkholderia schumannianae* UZHbot8) [33] to 11.5 Mb (*Burkholderia terrae* BS001), are characterized by a high G + C content (62–68 mol%) and consist of multiple replicons [34]. To further infer the genome-wide taxonomic affiliation of strains SRS-25 and SRS-46, EDGAR analysis was run (Figure 3). A genome-wide phylogenetic tree from this analysis revealed that both these strains cluster within the *Burkholderia cepacia* complex (Bcc); a group of approximately 20 closely related species [35]; many isolates have demonstrated commercial potential as biological control agents of plant pathogens, bioremediation of recalcitrant xenobiotics and plant growth promoting abilities. Moreover, the Bcc group is ubiquitously found ranging from soils, aquatic habitats, and plant rhizosphere niches, associated with humans, animals and also pathogens in hospital environments. Bcc species are extremely versatile metabolically by utilizing more than 200 organic compounds, are resistant to multiple antibiotics and even fix atmospheric nitrogen (N_2_). Strains from the Bcc group possess large-plastic genomes represented by multiple chromosomes- typically ranging from 2 to 4 replicons, which contributes to ecological competitiveness of the Bcc group.

Notably, the EDGAR analysis also revealed that strains 25, 46 and a previously isolated strain from our research (*Burkholderia* sp. strain SRS-W-2-2016), share a common pool of 3128 genes (Figure 4A). Moreover, several distinctive genes, shown in parenthesis, were also identified from each of the closest taxonomic relatives of the SRS strains (Figure 4B): *Burkholderia lata* strain *383* (1308); *B*. *multivorans* (1056); *B*. sp. strain SRS-25 (826), *B*. sp. strain SRS-46 (1749) and *B*. sp. strain SRS-W-2-2016 (2580), respectively. Among the suite of 3128 genes common only to the SRS strains, we found several metal resistant gene homologues as shown in Table 1. Of specific interest were genes that encoded metal resistance proteins (e.g., for cadmium, cobalt, and zinc), transporter proteins, stress proteins, cytochromes, and drug resistance, respectively.

### 3.3. Comparative Proteomics Study to Analyze Response to Uranium Exposure

Proteome profiles of *Burkholderia* sp. (SRS-25 and SRS-46) amended with U were compared with unamended controls at 24 h when maximum depletion of uranium was observed (Figure 1C). The proteomics experimental scheme is shown in Figure 5.

Label free quantification (LFQs) of the proteomic data is shown in Appendix A. When the obtained proteomes were visualized using a Venn diagram, it became apparent that a total of 276 and 408 proteins were common between strains SRS-25 and SRS-46 (Figure 6). Interestingly, proteins were found in different fold ratios between unamended and U amended conditions (Appendix A). Further analysis of these proteins showed that approximately 34 proteins in SRS-25 and 52 proteins in SRS-46 were (~2 or >2-fold change) found in abundance in the presence of U relative to the control. These proteins are listed (Table 2) and categorized according to their function to obtain a broader understanding on U resistance mechanisms. Most of these proteins were functionally categorized into protein biosynthesis, transport, damaged DNA repair and stress response; details are presented in the following sections with a focus on ~2-fold or higher proteins expressed in U microcosms relative to the controls.

### 3.4. Protein Biosynthesis and Growth

Several proteins which are involved in biosynthesis, (for e.g., Proline-tRNA ligase, Methionine-tRNA ligase) and translation (e.g., Elongation factor G1, Translation initiation factor IF-2) were upregulated under U stress, which suggests active protein synthesis as a mechanism for the bacterial cell to counter U stress. Interestingly, transporter proteins, such as protein translocase subunit SecA, which is a Zn binding metal protein, was also found to be upregulated in the presence of U (4.7 fold higher in SRS-25). The higher abundance of proteins from this category is consitent with earlier study [36], suggesting that cells grown in the presence of U were actively involved in transport, possibly of U. Moreover, the proteins involved in ribosome biogenesis and amino acid biosynthesis were also upregulated indicative of active cellular metabolism.

### 3.5. DNA Damage, Repair and Stress Response

It has been shown previously that U can bind to DNA and can cause DNA damage/breakage [37,38]. DNA ligase plays a major role in DNA replication and DNA repair and was found highly up-regulated (17 and 2.5-fold higher in SRS-25 and 46, respectively) in both the strains (Table 2) upon exposure to U. This likely seems a strong cellular mechanism to repair damage imposed by U toxicity. Additionally, Chaperone protein DnaK, involved in DNA replication and stress response, also showed 1.9 fold increase in SRS-25. Similarly, other DNA damage response proteins like Holliday junction ATP-dependent DNA helicase RuvA, and lexA repressor were also expressed in higher amount in SRS-46. The higher upregulation of these proteins suggests the possibility of DNA damage due to U exposure. Stress response proteins such as Chaperone protein HtpG, Chaperone protein HscA homolog, Co-chaperone protein HscB homolog, Lon protease etc. were also abundant in both SRS-25 and SRS-46 strains. These proteins are generally expressed in response to cellular stress to rectify misfolding of proteins [39], which was likely induced by U. The increased expression of peptide methionine sulfoxide reductase MsrB, having a role in oxidative stress and protein repair, further supports the above possibility. These responses are also in agreement with earlier studies [36], where U caused DNA damage and expression of heat labile proteins.

The expression of thymidylate kinase was also highly increased in SRS-46, which is involved in nucleotide biosynthesis; a higher expression of this protein may be due to the requirement of nulceotides during DNA repair process.

### 3.6. Metabolism and ATP Synthesis

The expression of N-succinylarginine dihydrolase, which is involved in Arginine metabolism and also in stress response was increased (3.2 fold) in SRS-46. Other proteins like Thymidine phosphorylase involved in pyrimidine metabolic process also increased (Table 2) along with proteins involved in ATP synthesis like NADH-quinone oxidoreductase subunit 1 and ATP synthase subunit delta. This is suggestive of the cellular machinery utilizing the produced ATP for increased metabolism response or even U detoxification.

### 3.7. Membrane Damage and Other Proteins

The higher abundance of LPS-assembly protein LptD upon U stress is suggestive of membrane synthesis, which may have been damaged by binding or U toxicity. Similarly, increased expression of lipoproteins upon U exposure has been shown in *Geobacter sulfurreducens* [36]. Proteins with oxidoreducatse activity such as Thiol:disulfide interchange protein DsbA were increased (Table 2), most likely due to the cells’ defense mechanism against oxidative stress. A CO_2_ fixation enzyme Phosphoenolpyruvate caboxylase was increased, probably to balance the redox potential of cell.

### 3.8. Uncharacterized Proteins

The expression of uncharacterized proteins in SRS-46 like UPF0234 protein Bmul_0741/BMUL1_02519, UPF0307 protein Bcep18194_A4194, UPF0301 protein Bamb_0737 was found higher in the presence of U. These hetherto uncharacterized proteins may be involved in conferring U resistance or providing some form of protection to the bacterial cells under U stress. Further research to fully characterize these proteins and establish their potential role(s) in U resistance is required to validate these speculations.

### 3.9. Statistical Analysis

Proteomics data was used to evaluate differences that may have ocurred in strains 25 and 46 as a function of uranium exposure relative to the controls. Preliminary analysis plotted as heatmaps (Appendix A), which revealed that U exposure resulted in the identification of a plethora of upregulated proteins in both strains relative to the non-uranium containing microcosms. For better visualization, we then selected the top 50 proteins based on the fold-change, from either strains, plotted to show distinctions observed between controls and uranium exposure; this is shown in Figure 7A–C.

## 4. Discussion

The ability of bacterial strains to persist in uraniferous ecosystems has been well documented for anaerobic microbiota, which are well known to be engaged in uranium cycling processes [36]. However, the potential of aerobic microorganisms to biomineralize uranium has been largely ignored and hence understudied. Interestingly, *Burkholderia*-like microorganisms have been demonstrated to comprise a significant proportion of the total uraniferous soil microbiota at the Integrated Field-Scale Subsurface Research Challenge Site (IFRC), Oak Ridge, Tennessee [15], as well as Rifle, Colorado [16]; yet, not much is known about the cellular and molecular response(s) of Burkholderiales against U toxicity. In this study, *Burkholderia* sp. strains 25 and 46 were isolated from uraniferous soils of the Tims Branch- Steed Pond site in SRS, which is characterized by histroical contamination with heavy metals due to nuclear weapons’ production activities. Micrcosoms established with these strains in the presence of externally added uranium showed rapid U depletetion within 24 h. There are at least four well-known mechanisms by which bacteria can transform and resist U toxicity. These include reductive precipitation by outer membrane cytochromes, pili, or spores; surface adsorption by exopolysaccharide (EPS) or S-layers; or biomineralization of U with the help of phosphatase enzymes [40,41]. A phosphatase positive response of both strains on TPMG media indicates their ability to biomineralize uranium, but this needs to be rigorously tested.

To have a better understanding of the metabolic potential possessed by the two *Burkholderia* strains, draft genome sequences were prepared and analyzed, which resulted in the identification of a plethora of genes related to metal resistance, transporter proteins, cytochromes and drug resistance (Table 1). The genomics analysis confirms the strong metabolic and bioremediation potential possessed by these strains that likely facilitates survival in the co-contaminated SRS soil habitat. As stated above, *Burkholderia* spp. has previously been shown to successfully colonize habitats characterized with extreme environmental stresses, including uraniferous niches; this study points to an arsenal of genome-enabled traits to do so successfully.

Further, which of the genome-enbled mechanisms mechanism(s) were upregulated due to uranium exposure were further probed by evaluating proteomic changes in the presence or absence of uranium stress. Note that despite predominance of *Burkholderia* spp., documented at several uranium contaminated environments, studies have not been performed to assess how these Proteobacterial members respond against uranium stress, which could provide critical clues on uranium bioremediation and recommendations on better stewardship of nuclear materials contaminated ecosystems can be offered. To our knowledge, there are only a few studies available, where different bacterial strains were exposed to U and proteomes were analyzed [19,20,36,42,43].

To address this knowledge gap, specifically in context to *Burkholderia* species, our proteomics-guided approach led to the identification of 276 and 408 commonly expressed proteins between strains SRS-25 and SRS-46 (Figure 6). Arguably, the number of proteins detected in this study were found to be unusually low and this occurs typically from poor sample preparation and LC discrimination. However, the raw data obtained, such as the LC control runs (Appendix A) and the number of solubilized membrane proteins (Appendix A) do not support this notion. Yet another reason that may likely have an impact on the proteome analysis presented in this study could be the use of a publicly available *Burkholderia* proteome database via UniProt. In the event that strains SRS-25 and SRS-46 contained large genomic deviations from the publicly available database, it may cause a significant impact on the proteome search. However, this also does not appear to be the case by taxonomic analysis of the draft genomes of strains 25 and 46 (Figure 3), which shows genome-wide affiliations of strain 25 with *Burkholderia lata* and strain 46 with *Burkholderia multivorans*, respectively. Regardless, this discrepancy can likely be addressed in our future work in which we will use translated genomes of strains SRS-25 and SRS-46 and then run proteomics comparisons. We, however, strongly believe that searching the proteomes from translated databases of environmental *Burkholderia* strains would have limited value because comparisons with the entire *Burkholderia* database will reveal better proteomic insights mainly due to rigorous annotations and functional evaluations of the known proteins from different experimental studies as opposed to using just one or two strains isolated from uraniferous soils.

Despite low protein yeilds, this study led to the identification of a set of upregulated proteins as a function of uranium exposure—34 proteins in SRS-25 and 52 in SRS-46, respectively (Table 2). Furthermore, statistical analysis indicated that each of the two strains possess their unique set of proteins to counteract stress and toxicity imposed by uranium, as a shared pool of proteins being expressed in both strains due to uranium exposure was not clearly evident (Figure 7). Despite the fact that both strains were isolated from the same soil and season, yet their total proteomics response against uranium toxicity seemed to be counteruntuitive, such that the proteomes of the two strains differed but not as a function of uranium alone. It may be that the *Burkholderia* strains 25 and 46 colonized different microniches in the soils from where they were isolated and thus, were exposed to different levels of bioavailable uranium, and hence, evolved different proteomic responses against uranium toxicity. Moreover, future protein validations will provide further evidence to completely understand why proteomes of the two *Burkholderia* species were statistically different with uranium not exhibiting the expected impact (Figure 7), and more importantly, identification of uranium-specific protein targets in Burkholderiales.

Regardless, upon further review, the unique 34 proteins from strain SRS-25 and 52 from SRS-46, were those that were previously shown to function in bacterial stress response, DNA repair, protein biosynthesis, and metabolism. By inference, it appeared that uranium upregulated multiple stress response pathways and heavy metal resistance functions to enable ecological and evolutionary survival in uraniferous soils. Multiple stress response pathways became functional in the two *Burkholderia* strains most likely due to their innate ability to resist higher concentrations of uranium, as shown in this study (Figure 1). In contrast to our study, previous proteomics studies have been performed at much lower concentrations of uranium that do not represent environmentally relevant concentrations, especially in context to the SRS soil habitat. For example, *Caulobacter crescentus* was exposed to 200 µM or 500 µM of uranyl nitrate (Yung et al. 2014); *Geobacter sulfurreducens* at 100 µM of uranyl acetate [36]; *Acidithiobacillus ferrooxidans* to 0.5 mM of uranium [42]; the nitrogen fixing strain, *Anabaena*, was exposed to 75 µM and 200 µM of uranyl carbonate [42] and *Microbacterium oleivorans* A9 at 10 µM of uranyl nitrate [19], respectively. Notably, a recent study from our laboratory suggests that bioavailable uranium is approximately 5 to 30 times lower than the total soil uranium concentrations (~4.2 mM) that is typically been documented from the historically contaminated SRS soils [22]; thus, we plan to repeat the proteomics studies at environmentally bioavailable concentrations of uranium, results of which will further improve our basic understanding on uranium-bacterial interactions at the cellular and molecular level.

Noteworthy also, were the upregulation of previously uncharacterized proteins in SRS-46, such as the UPF0234 protein Bmul_0741/BMUL1_02519, UPF0307 protein Bcep18194_A4194, and UPF0301 protein Bamb_0737, which were higher in the presence of uranium. Other studies on *Geobacter* sp., *Microbacterium* sp., *Acidithiobacillus* sp., have also led to the identification of uncaracterized proteins under uranium stress [19,42], and thus, points to the lack of knowledge on the underpinnings of uranium-microbial interactions at the molecular level. It is very likely that the suite of uncharacterized proteins may also be engaged in conferring resistance or even biomineralization of uranium in Burkholderiales, which warrants further research.

In summation, this study further enhances our understanding of the multiple proteogenomic mechanisms of two newly isolated and soil-borne *Burkholderia* strains against uranium stress and provides for a framework for additional studies, especially at much shorter time-scales (e.g., hourly intervals post uranium exposure), and more importantly, at bioavailable and environmentally relevant concentrations of uranium, such that precise molecular mechanism(s) underpinning *Burkholderia*-uranium interactions and remediation can be better understood.

## Figures and Tables

**Figure 1 cells-07-00269-f001:**
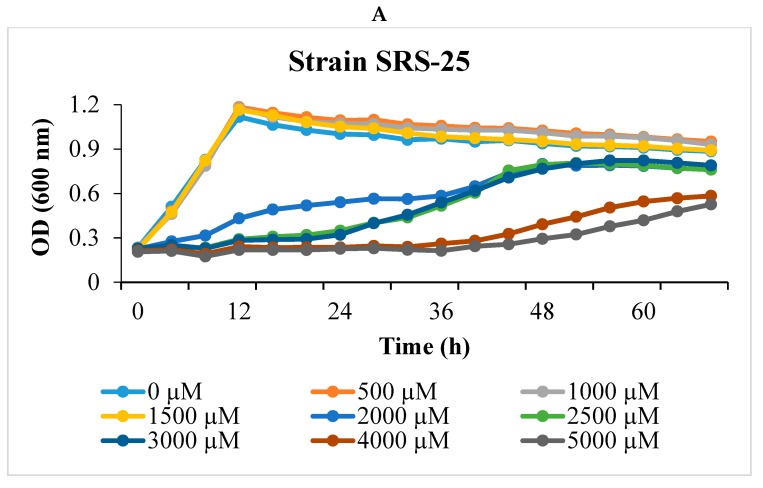
Shown is the resistance abilities of *Burkholderia* sp. SRS-25 (**A**) and SRS-46 (**B**) at uranium concentration ranging from 0 to 5000 µM. Also shown is the growth (Bioscreen C) and concomitant depletion of uranium evaluated by inductively coupled plasma-mass spectrometry (ICP-MS) in microcosms spiked with 1000 µM uranium (**C**); red lines depict growth at OD_600_ strain along with depletion of uranium shown in blue color, respectively.

**Figure 2 cells-07-00269-f002:**
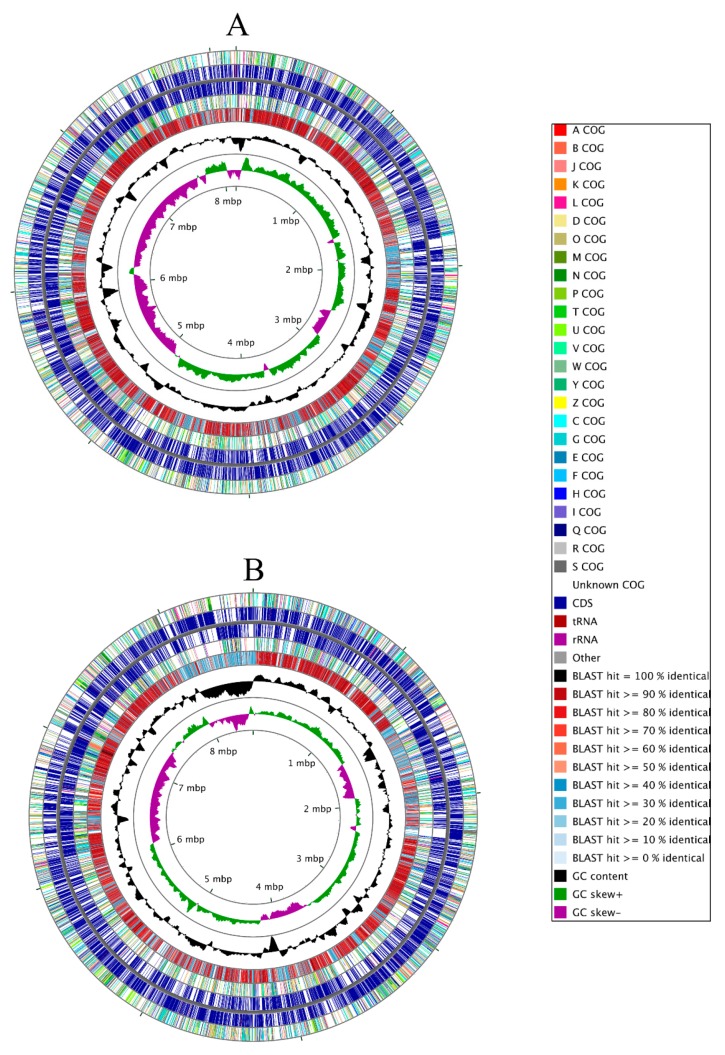
Shown are circular genomic maps of *Burkholderia* sp. strain SRS-25 (**A**) and SRS-46 (**B**). The first (outermost) and fourth rings depict COG categories of protein coding genes on the forward and reverse strands; the second and third rings show the locations of protein coding, tRNA, and rRNA genes on the forward and reverse strands; the fifth ring displays regions of similarity detected using BLAST (E-value = 0.0001) between strain SRS-25 or SRS-46 coding sequence translations and those from the genome of *Burkholderia multivorans* ATCC 17616 (accessions CP000868, CP000869, CP000870, CP000871). Regions of similarity are colored based on the percent identity between the aligned proteins. The black plot depicts GC content with the peaks extending towards the outside of the circle representing GC content above the genome average, whereas those extending towards the center mark segments with GC content lower than the genome average. The innermost plot depicts GC skew. Both base composition plots were generated using a sliding window of 50,000 nt.

**Figure 3 cells-07-00269-f003:**
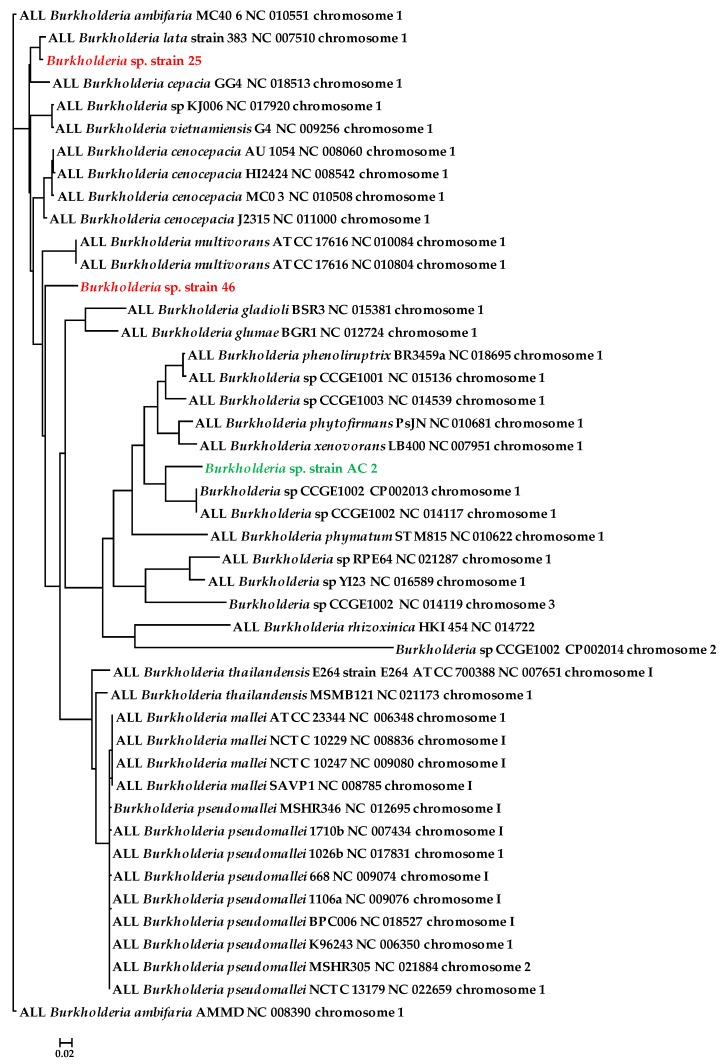
Depicted is the genome-wide phylogenetic tree of *Burkholderia* species reported in this study. Tree for 45 genomes was built using the EDGAR pipeline, out of a core of 2 genes per genome, 90 in total. The core consists of 1155 AA-residues/ bp per genome, 51,975 in total. Shown in red color are the two *Burkholderia* species isolated from SRS soils reported in this study, and the one shown in green color was obtained from a previous study.

**Figure 4 cells-07-00269-f004:**
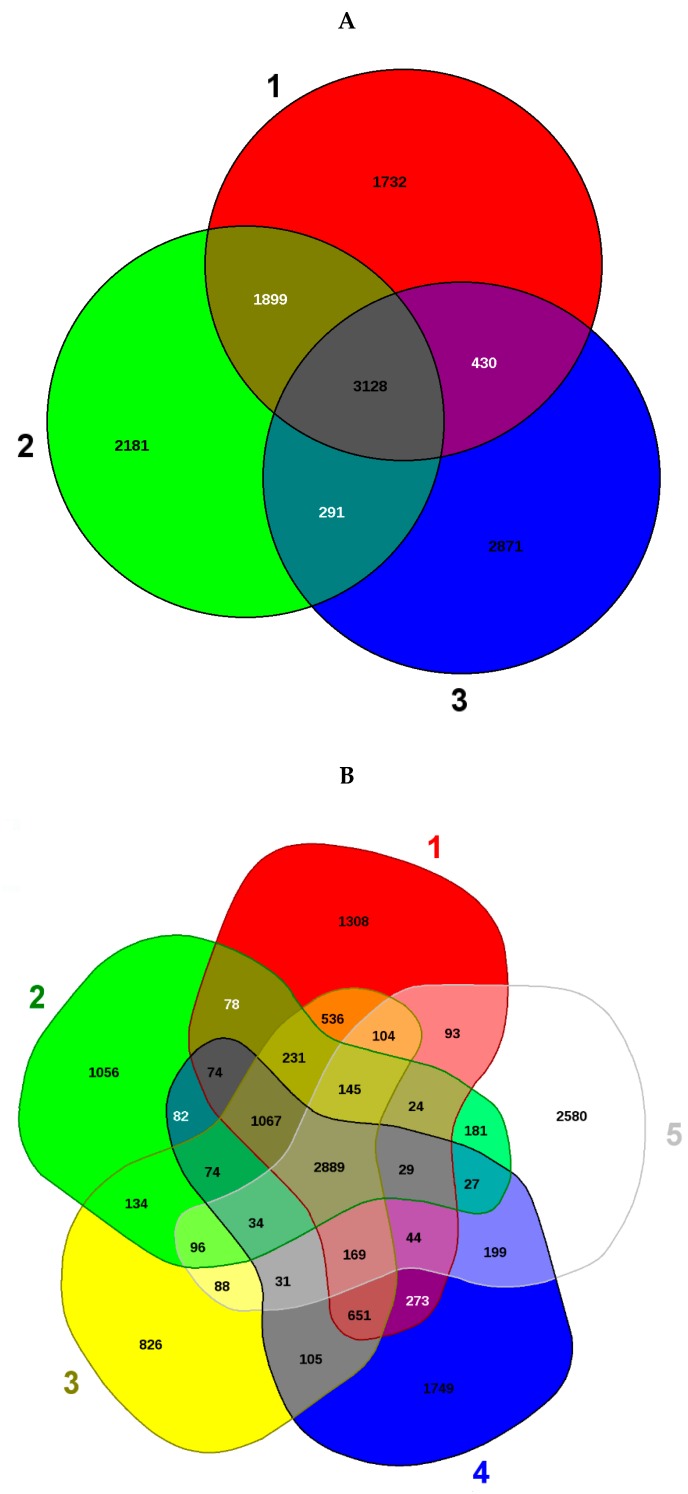
Shown are the whole-genome sequence-based Venn diagrams generated by the EDGAR pipeline revealing strains SRS-25 and 46 along with a previously isolated strain from our research (*Burkholderia* sp. strain SRS-W-2-2016). These isolates share a common pool of 3128 genes (**A**); Venn diagram sectors belong to 1 (red), Strain SRS-25; 2, SRS-46 (green) and 3, SRS-W-2-2016 (blue), respectively. Shown in (**B**) are several distinctive genes identified when comparisons were run between the two closest taxonomic relatives: *Burkholderia lata* strain 383 (sector 1, red); *B*. *multivorans* (sector 2, green); strain SRS-25 (sector 3, yellow); strain SRS-46 (sector 4, blue) and SRS-W-2-2016 (sector 5, white). The core genomes are shown in the centered gray area.

**Figure 5 cells-07-00269-f005:**
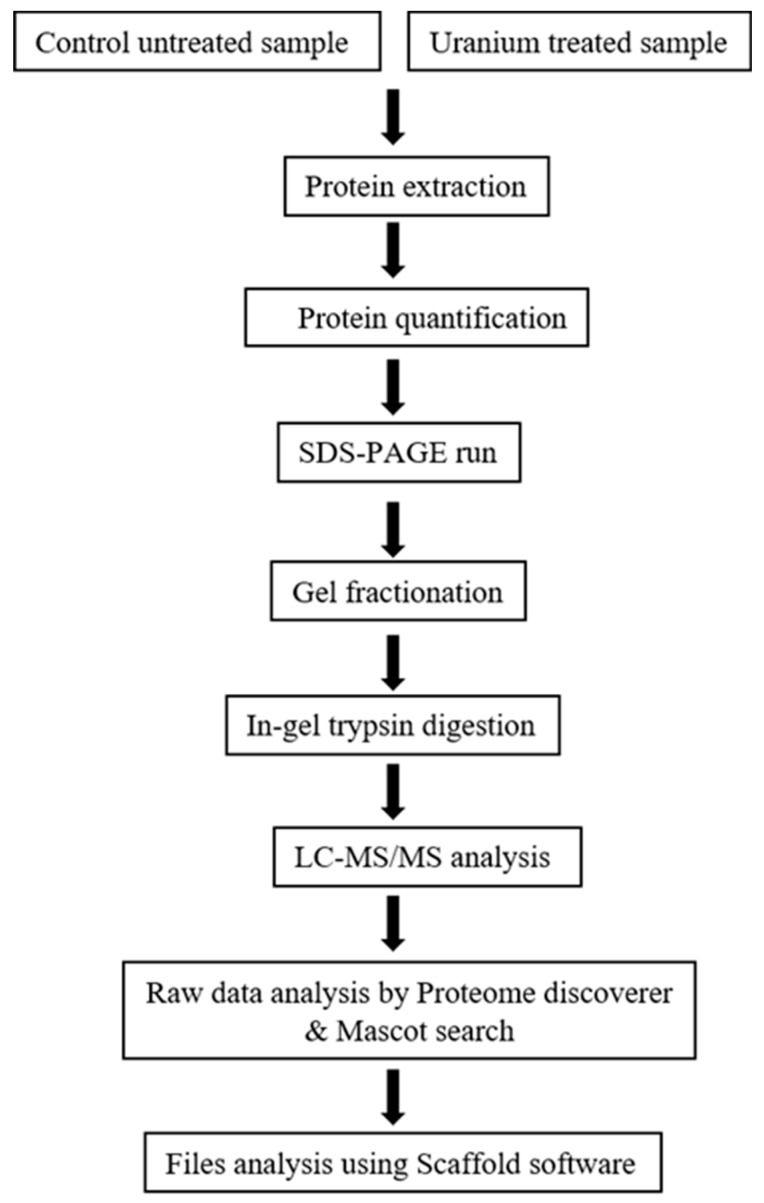
Shown is the proteomics workflow followed in this study.

**Figure 6 cells-07-00269-f006:**
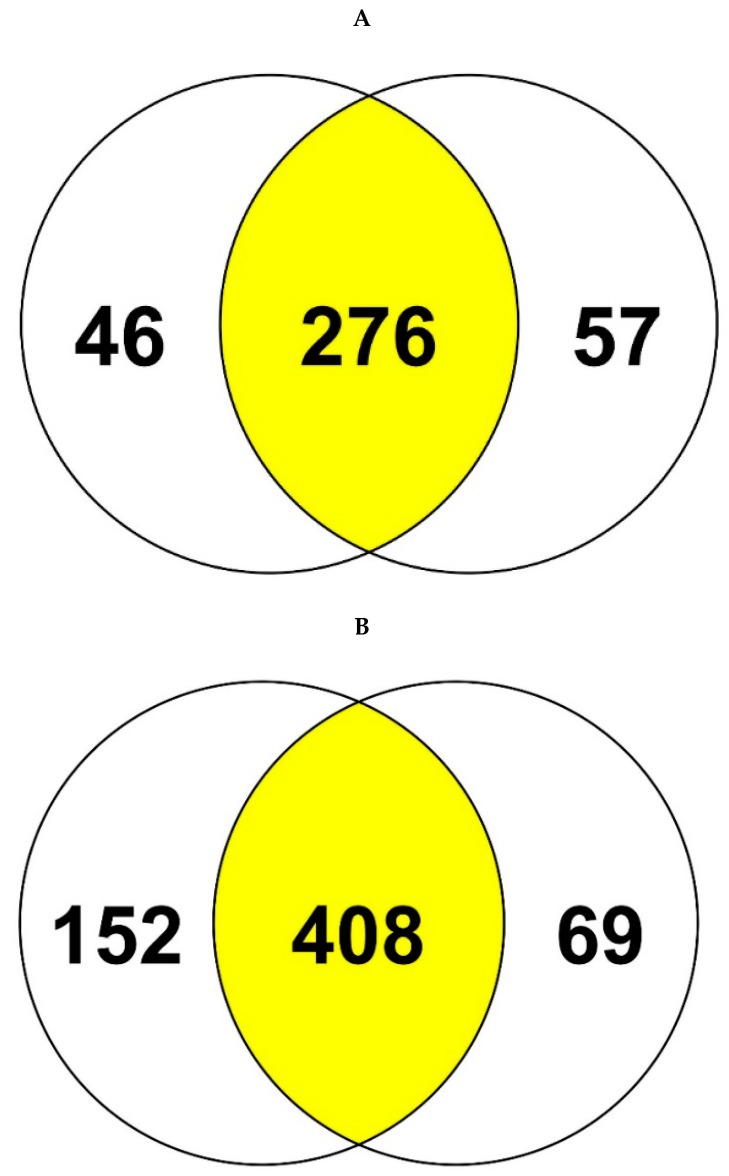
Proteomics data plotted as a Venn diagrams from (**A**) *Burkholderia* sp. strain SRS-25 and (**B**) *Burkholderia* sp. strain SRS-46 (**B**), respectively. Venn diagram sectors represent proteins expressed in control (left circle) and uranium treated (right circle). The yellow segment represents the number of commonly expressed proteins between untreated and U treated samples.

**Figure 7 cells-07-00269-f007:**
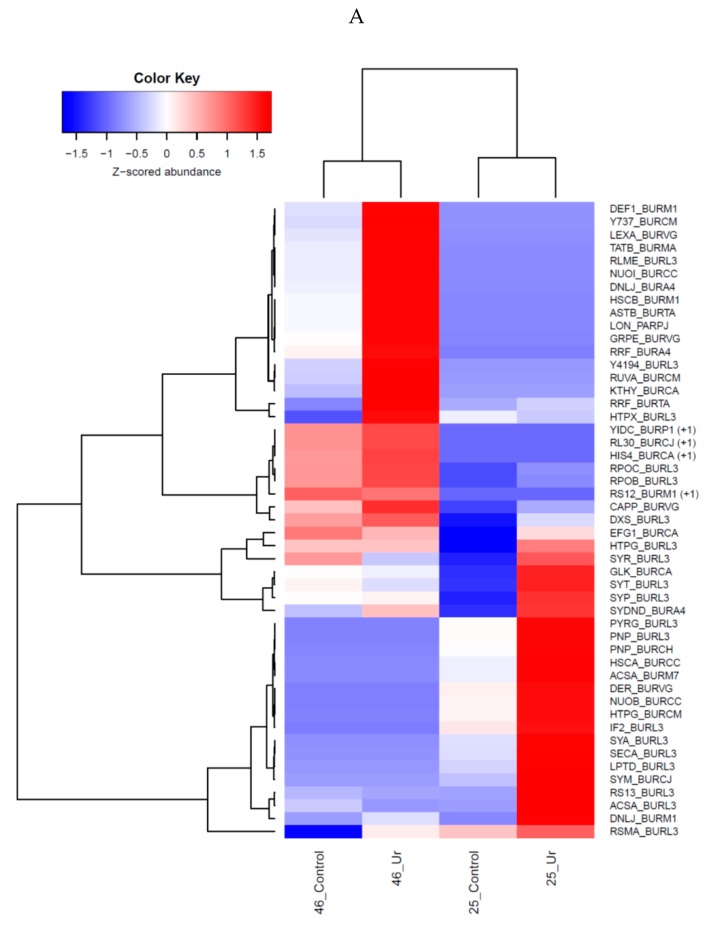
Shown is the heatmap obtained from *Burkholderia* sp. strains 25 and 46 with top 50 proteins by fold-change from either of the two strains plotted to show their distinctions observed between controls and uranium exposure (**A**). Gradient of fold change is shown by color code, where blue represents the lower most fold change value and red represents the highest fold change value. Parenthesis in yellow and green color represents those proteins that were identified as over-represented in strains 25 and 46; (**B**) Nonmetric multidimensional scaling (NMDS) ordination of all identified proteins from either strain; Bray-Curtis similarity values between are shown at the 40%, 60%, and 80% levels; and (**C**) Dendrogram based cluster analysis of the NMDS of total identified proteins. Data was standardized, transformed by log (X + 1) and resemblance matrix was calculated prior to plotting the NMDS and dendrogram.

**Table 1 cells-07-00269-t001:** Shown are the EDGAR identified core set of gene homologues common between *Burkholderia* sp. strain SRS-25, strain SRS-46 and SRS-W-2-2016 that likely perform a biodegradative or metal resistance function.

Category	Gene Homologue
Transporter proteins	Proline/Betaine transporter
	MFS-type transporter YhjX, YcaD
	Phospholipid ABC transporter permease protein
	ABC transporter ATP-binding protein YbhF, YheS
	Cystine inner membrane transporter
	Hemolysin transporter protein ShlB precursor
	Divalent metal cation transporter MntH
	Inner membrane ABC transporter permease protein
	Riboflavin transporter
	Dicarboxylic acid transporter DauA
	Inner membrane transporter yiJE, YedA, YnfM
	Sulfoacetate transporter SauU
	Citrate transporter
	Lysophospholipid transporter LplT
	Niacin/nicotinamide transporter NaiP
	(3-hydroxy-phenyl) propionate transporter
	Tartrate transporter
	H(+)/Cl(−) exchange transporter ClcA
	d-galactonate transporter
	Low-affinity inorganic phosphate transporter 1
	l-galactonate (Hexuronate) transporter
	Heme/hemopexin transporter protein HuxB precursor
	Sialic acid transporter
	Amino-acid permease protein YxeN
	Glutamate/aspartate transporter permease protein
	Fluoride ion transporter CrcB
	Sialic acid transporter
	Fe (3+) ions import ATP-binding protein FbpC
	4-hydroxybenzoate transporter PcaK
	Glucarate transporter
	Efflux pump membrane transporter BepE
	Nitrate/nitrite transporter NarK2
	Lactose transport system permease protein LacF
	Manganese ABC transporter substrate-binding lipoprotein
	Uric acid transporter UacT
	Glutamine ABC transporter permease protein GlnM
	High-affinity gluconate transporter
	Phospholipid ABC transporter-binding protein MlaD
Stress proteins	Stress response kinase
	General stress protein 39, 69
	Universal stress protein
	Persistence and stress-resistance antitoxin PasI
	Persistence and stress-resistance toxin PasT
	TRAP-T-associated universal stress protein TeaD
	Acid stress protein IbaG
Cytochromes	Cytochrome c oxidase subunit 2 precursor
	Cytochrome c oxidase subunit 1
	Cytochrome c oxidase assembly protein CtaG
	Cytochrome c oxidase subunit 3
	Cytochrome bd ubiquinol oxidase subunit X
	Cytochrome bd-I ubiquinol oxidase subunit 2
	Cytochrome bd ubiquinol oxidase subunit 1
	Gluconate 2-dehydrogenase cytochrome c subunit precursor
	Cytochrome c4 precursor
	Cytochrome b561
	Succinate dehydrogenase cytochrome b556 subunit
	Cytochrome c-552 precursor
	Cytochrome c-554(548)
	Quinone-reactive Ni/Fe-hydrogenase B-type cytochrome subunit
	Fructose dehydrogenase cytochrome subunit precursor
	Sulfide dehydrogenase [flavocytochrome c] flavoprotein chain precursor
	Cytochrome bo (3) ubiquinol oxidase subunit 4
	Cytochrome bo (3) ubiquinol oxidase subunit 3
	Cytochrome bo (3) ubiquinol oxidase subunit 2 precursor
	Cytochrome c biogenesis protein CcsA
	Cytochrome c1 precursor
	Cytochrome b/c1
	Ubiquinol-cytochrome c reductase iron-sulfur
	Cytochrome c biogenesis protein CcsB
	Cytochrome c-555 precursor
	Cytochrome b556(fdo) subunit
Metal resistance proteins	ATM1-type heavy metal exporter
	Divalent metal cation transporter MntH
	ATP-dependent zinc metalloprotease FtsH
	Metal-dependent hydrolase YcfH, YjjV
	Metallo-beta-lactamase type 2
	Metal chaperone YciC
	Metallo-hydrolase YycJ
	Metalloprotease PmbA
	High-affinity nickel transport protein
	Nickel and cobalt resistance protein CnrA
	Nickel-binding periplasmic protein
	Magnesium and cobalt efflux protein CorC
	Cobalt-zinc-cadmium resistance protein CzcA, CzcC, CzcB
	Cobalt/magnesium transport protein CorA
	Copper resistance protein C precursor
	Copper-exporting P-type ATPase A
	Copper homeostasis protein CutC
	Arsenical resistance operon repressor
	Arsenical-resistance protein Acr3
	Manganese ABC transporter substrate-binding lipoprotein
	Manganese transport system membrane protein MntB
	ATP-dependent zinc metalloprotease FtsH
	Zinc-type alcohol dehydrogenase-like protein
	Zinc uptake regulation protein
	Zinc import ATP-binding protein ZnuC
	High-affinity zinc uptake system membrane protein ZnuB
	Zinc transport protein ZntB
	Cadmium-transporting ATPase
Drug resistance	Multidrug resistance protein MdtH, MdtE, MdtC, NorM, stp
	Multidrug export protein EmrB, EmrA
	Multidrug resistance outer membrane protein MdtP
	Multidrug efflux pump subunit AcrB
	Multidrug resistance protein 3, EmrK
	Multidrug export ATP-binding/permease protein

**Table 2 cells-07-00269-t002:** List of upregulated proteins in SRS-25 and SRS-46 strains, following exposure to uranium. Fold change represents the expression with U amendment normalized to expression without U, respectively.

Function	Protein	Gene	Fold Change	Strain
Protein synthesis, translation and transport	Aspartate-tRNA(Asp/Asn) ligase	aspS	2.3	SRS-25
	Alanine-tRNA ligase	alaS	4.5	SRS-25
	Elongation factor G1	fusA1	2.4	SRS-25
	Proline-tRNA ligase	proS	5.6	SRS-25
	Arginine-tRNA ligase	argS	3.8	SRS-25
	Glutamate-tRNA ligase	gltX	2	SRS-25
	Methionine-tRNA ligase	metG	9.6	SRS-25
	Translation initiation factor IF-2	infB	2.3	SRS-25
	Elongation factor G2	fusA2	2.1	SRS-25
	Protein translocase subunit SecA	secA	4.7	SRS-25
	Threonine-tRNA ligase	thrS	5.2	SRS-25
	50S ribosomal protein L20	rplT	1.9	SRS-25
	50S ribosomal protein L10	rplJ	1.7	SRS-25
	30S ribosomal protein S13	rpsM	3.2	SRS-25
	30S ribosomal protein S7	rpsG	1.9	SRS-46
	30S ribosomal protein S5	rpsE	1.8	SRS-46
	50S ribosomal protein L5	rplE	1.8	SRS-46
	Ribosome-recycling factor	frr	2.7	SRS-46
	50S ribosomal protein L25	rplY	1.8	SRS-46
	NADH-quinone oxidoreductase subunit C	nuoC	2	SRS-46
	Sec-independent protein translocase protein tatB	tatB	3.8	SRS-46
	Peptide deformylase 1	def1	4.7	SRS-46
	Ribosomal RNA small subunit methyltransferase A	rsmA	3.8	SRS-46
	Ribosomal RNA large subunit methyltransferase E	rlmE	3.8	SRS-46
Electron transport coupled proton transport	NADH-quinone oxidoreductase subunit B	nuoB	2.5	SRS-25
Ribosome biogenesis	GTPase Der	der	2.5	SRS-25
	30S ribosomal protein S13	rpsM	3.2	SRS-25
	Endoribonuclease YbeY	ybeY	2.5	SRS-46
Amino acid biosynthesis	Dihydroxy-acid dehydratase 1	ilvD1	2.3	SRS-25
	Argininosuccinate lyase	argH	1.8	SRS-46
	Imidazole glycerol phosphate synthase subunit HisF	hisF	1.7	SRS-46
	3-isopropylmalate dehydratase small subunit	leuD	1.8	SRS-46
	Homoserine O-succinyltransferase	metXS	1.7	SRS-46
	Imidazole glycerol phosphate synthase subunit HisH	hisH	2.5	SRS-46
Transcription	DNA-directed RNA polymerase subunit beta	rpoB	3.8	SRS-25
	DNA-directed RNA polymerase subunit beta	rpoC	2.9	SRS-25
	Bifunctional protein glk	glk	3.1	SRS-25
DNA Replication, recombination and repair	Chaperone protein DnaK	dnaK	1.9	SRS-25
	DNA ligase	ligA	17	SRS-25
	DNA ligase	ligA	2.5	SRS-46
	lexA repressor	lexA	4.2	SRS-46
	Holiday junction ATP-dependent DNA helicase RuvA	ruvA	7	SRS-46
Stress response	Chaperone protein HtpG	htpG	2.6	SRS-25
	Chaperone protein HscA homolog	hscA	3.5	SRS-25
	Polyribonucleotide nucleotidyltransferase	pnp	2.9	SRS-25
	N-succinylglutamate 5-semialdehyde dehydrogenase	astD	2.4	SRS-25
	Lon protease	lon	3.2	SRS-46
	Protease HtpC homolog	htpX	3.1	SRS-46
	Protein GrpE	grpE	3	SRS-46
	Co-chaperone protein HscB homolog	hscB	3.2	SRS-46
	60 kDa chaperonin 2	groL2	1.7	SRS-46
	Chaperone protein DnaJ	dnaJ	1.7	SRS-46
	Heat-inducible transcription repressor HrcA	hrcA	1.9	SRS-46
Nucleotide biosynthesis	CTP synthase	pyrG	2.8	SRS-25
	Thymidylate kinase	tmk	11	SRS-46
Isoprene and thiamine biosynthesis	1-deoxy-D-xylulose-5-phosphate synthase	dxs	5.4	SRS-25
	2-C-methyl-D-erythritol 2,4-cyclodiphosphate synthase	ispF	2.1	SRS-25
CO_2_ fixation	Phosphoenolpyruvate carboxylase	ppc	3.6	SRS-25
	Phosphoenolpyruvate carboxylase	ppc	1.9	SRS-46
Outer membrane assembly protein, lipopolysaccharide transport	LPS-assembly protein LptD	lptD	5.8	SRS-25
Oxidative stress response and protein repair	Peptide methionine sulfoxide reductase MsrB	msrB	1.9	SRS-25
Metabolism	Acetyl-coenzyme A synthetase	acsA	3.6	SRS-25
	Enolase	eno	1.7	SRS-46
	N-succinylarginine dihydrolase	astB	3.2	SRS-46
	Thymidine phosphorylase	deoA	2.5	SRS-46
	Phophoglucosamine mutase	glmM	1.7	SRS-46
	Anhydro-N-acetylmuramic acid kinase	anmk	2.5	SRS-46
	dCTP deaminase	dcd	1.7	SRS-46
Nucleotide biosynthesis	Thymidylate kinase	tmk	11	SRS-46
Oxidoreductase activity and antibiotic response	Thiol:disulfide interchange protein DsbA	dsbA	2.2	SRS-46
Ion channnel, ion transport	Large-conductance mechanosensitive channel	mscL	2.5	SRS-46
Aerobic respiration, ATP synthesis	NADH-quinone oxidoreductase subunit 1	nuol	3.8	SRS-46
	ATP synthase subunit delta	atpH	1.9	SRS-46
Kinase, transferase	Cytidylate kinase	cmk	1.7	SRS-46
Lipid biosynthesis	3-hydroxy-[acyl-carrier-protein]	fabZ	1.9	SRS-46
	Lipid-A-disaccharide synthase	lpxB	1.9	SRS-46
Pyridoxine biosynthesis	4-hydroxythreonine-4-phosphate dehydrogenase	pdxA	2.5	SRS-46
Queuosine biosynthesis	7-carboxy-7-deazaguanine synthase	queE	2.5	SRS-46
Ubiquinone biosynthesis	2-nonaprenyl-3-methyl-6-methoxy-1,4 benzoquinol hydroxylase	coq7	1.7	SRS-46
Glycosyl transferase	Uracil phosphoribosyltransferase	upp	2.1	SRS-46
Poorly characterized	UPF0234 protein Bmul_0741/BMUL1_02519	Bmul_0741	1.9	SRS-46
	UPF0307 protein Bcep18194_A4194	Bcep18194_A4194	6.4	SRS-46
	Probable transcriptional regulatory protein Bphyt_1301	Bphyt_1301	2.1	SRS-46
	UPF0234 protein Bphy_0527	Bphy_0527	1.7	SRS-46
	Probable transcriptional regulatory protein Bamb_2332	Bamb_2332	2.3	SRS-46
	UPF0301 protein Bamb_0737	Bamb_0737	5.1	SRS-46

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
