# Peer review of "Proteogenomic Analysis of Burkholderia Species Strains 25 and 46 Isolated from Uraniferous Soils Reveals Multiple Mechanisms to Cope with Uranium Stress"

_cells, 2018, doi:10.3390/cells7120269_

Round 1
Reviewer 1 Report
The manuscript “Proteogenomic Analysis on Burkholderia species Strains 25 and 46 Isolated from Uraniferous Soils Reveals Multiple Mechanisms to Cope with Uranium Stress” by Meenakshi Agarwal and co-workers presents the isolation, genomic and proteomic analysis of 2 strains form U contaminated soil. Overall the manuscript is well written and easy to follow. There are, in my opinion, some questions that should be addressed:
1) In the tables 1 and 2 - i) The legend says over expressed proteins. Proteins are not expressed, please use accumulation or decrease in quantity instead. ii) Why didn’t the authors include the proteins that accumulated exclusively upon U amendment (in the venn diagrams in the figure above). Since in line 389 – 390 the authors state that the “bioremediative and/or adaptive mechanisms are utilized by strains SRS-25 and SRS-46 to circumvent U toxicity and stress remains unknown at this time”, some hint may be given by the inclusion of this set of data in the analyses iii) Minor misspells in some protein manes in the tables.
2) Figure 1C – there is an increase in the U detected by ICP in the strain 25 after 25 hours, which, i fell should be discussed in the text.
3) The figures 1 A and 1B and supplementary figure S2 must be improved for their quality is very low.
4) In lines 389-390 the authors state that the mechanism of yet in previously in the text (line 350) the authors mention that “stress is suggestive of membrane synthesis, which may have been damaged by binding or U toxicity”. The membrane damage, the presence of ESM can probably easy to spot using microscopy.
5) Lines 420 – The authors state that “Regardless, this discrepancy can easily be resolved by using the translated genomes of strains SRS-25 and SRS-46 and then run the proteomics comparisons.” Why was this strategy not implemented, or will it be included in future experiments with different U concentration. This gives the felling of unfinished work.
Minor issues
Lines 120 - the information provided does not match the data on the graphs. There is probably misspell. Some other misspells are in tables (names of proteins) and thoughtout the text (e.g. line 465 “...and necessitates the need ...”
The authors should mention in the materials and methods section the time point at which the protein extraction was preformed.
My best regards
Author Response
Please see attached response in word document.

Reviewer 2 Report
In this manuscript, authors isolated two strains (SRS-25 and SRS-46) of Burkholderia spp and checked the depletion/resistance of Uranium by both strains. After determining the resistance of uranium, authors not only performed a genomic analysis but also executed a proteomics study to evaluate the response imposed by uranium. The rationale behind the comparative proteomics study was comprehensible that the authors were interested to check the differential expression of various important proteins altered by uranium in the environment. However, the way in which the samples were prepared for the proteomics study is not clear. In the method section, authors solubilized the cells in SDT buffer (4% SDS) and lysed the cells by freeze-thaw methods. Afterward, authors centrifuged the lysed cells. It is not clear in the current version that authors run the only supernatant in SDS page or run pellet too. What happened with pellet because many membrane proteins are not soluble in 4% SDS. Either you have to resuspend the pellet in Urea/GdMcl or use different concentrations of various detergents to solubilize the membrane proteins. Authors should show not only the SDS gels displaying supernatant and pellet side by side but also depict the labeled fractions in SDS gels used for the proteomics study in the current study.
In general, the authors should explain how did they solubilize the membrane proteins and cut them from the gels for the peptide identification in SDS gels?
This imperfection can be the major reason that authors have poor protein yield for LCMS. How many membrane proteins in both strains were identified in control vs uranium treated samples? It is tough to envision that both strains from the same Burkholderia spp have expressed different genes under the same uranium treatment. Authors should show an SDS page showing an equal amount of proteins from both strains (SRS-25 and SRS-46) in both conditions, treated vs control samples.
Minor comments:
Figure 1 is crowded. Authors should remake this figure by simplifying the color scheme and X/Y axis. Authors should explain why did they prefer 1000um instead of 1500um? In figure1, both concentrations have similar behavior.
Authors can present a flow diagram showing the method of proteomics study?
Can the authors combine table 2 and table3? The sole purpose of these tables is that both stains express different genes under the same COG category. The combined table will be more biologically informative to evaluate the different lifestyle of both stains under same uranium treatment.
Figure 6 B and 6C are not clear. The authors should explain the rationale of these statistical graphs in term of strain lifestyle under the same uranium treatment.
In figure 6A, arrows are not shown.
The last paragraph of the discussion section is not clear. Authors should rewrite that section.
Please check the consistency of citations. They are missing in many places.
Page24, line 363, authors should move this “ statistical analysis ” to the method section.
Page 2, line 356. Are these genes “poorly characterized”? at what level? or these genes are hypothetical. Please rewrite this section.
Author Response

(The authors gave the same response as above.)

Round 2
Reviewer 2 Report
The current version of this manuscript is acceptable for the publication